# Exploring Trusted Sources of HPV Vaccine Information Among Mexican American Parents in El Paso, Texas

**DOI:** 10.3390/ijerph22010069

**Published:** 2025-01-07

**Authors:** Alyssa A. Martinez, Michelle Gil, Surendranath S. Shastri, Gabriel A. Frietze

**Affiliations:** 1College of Health Sciences, University of Texas at El Paso, El Paso, TX 79968, USA; aamartinez18@utep.edu (A.A.M.); mgil@miners.utep.edu (M.G.); 2Department of Health Disparities Research, Division of Cancer Prevention and Population Sciences, The University of Texas MD Anderson Cancer Center, Houston, TX 77030, USA; ssshastri@mdanderson.org; 3Department of Pharmaceutical Sciences, School of Pharmacy, University of Texas at El Paso, El Paso, TX 79968, USA

**Keywords:** human papillomavirus, HPV vaccine, Hispanic, trusted messengers, trusted sources of information

## Abstract

Hispanic populations are disproportionately impacted by HPV-associated cancers. An HPV vaccine is available that protects against 90% of HPV-associated cancers. Understanding the factors associated with HPV vaccine uptake, including identifying whom individuals trust to recommend the HPV vaccine, is an important step toward developing public health interventions for promoting the HPV vaccine among Hispanic people. The purpose of this pilot study was to use a qualitative approach to identify trustworthy messengers to disseminate HPV vaccine information among Mexican American (MA) parents of children of 11–17 years of age. Three 90 min pilot focus groups with three to five participants in each group were conducted. The inclusion criteria included being 18 years of age or older, residing in El Paso, TX, identifying as MA, speaking English, and being a parent of a child between the ages of 11 and 17. Focus groups were conducted with 15 predominately female participants (*M*_age_ = 38.46, *SD* = 5.73; Female = 93.3%). A reoccurring theme throughout all three focus groups was that pediatricians, registered nurses, and pharmacists were identified as the most trusted sources of information. Findings from this study have implications for designing public health interventions that leverage pediatricians, registered nurses, and pharmacists to promote the HPV vaccine among parents.

## 1. Introduction

Human papillomavirus (HPV) is the most common sexually transmitted infection (STI) in the United States, and Hispanic individuals are disproportionately impacted by HPV-associated cancers, including oropharyngeal and cervical cancer [1]. An HPV vaccine is available that protects against 90% of HPV-associated cancers and protects against nine types of HPV [1]. The CDC recommends the HPV vaccine for individuals aged 9–45 [1]. Only two doses of the vaccine are needed if an individual is vaccinated prior to the age of 15; however, three doses are needed for individuals between the ages of 15 and 45 [1]. Understanding the factors associated with HPV vaccine uptake, including identifying whom individuals trust to recommend the HPV vaccine, is an important step toward developing public health interventions for promoting the HPV vaccine.

Several studies have investigated trusted sources of HPV vaccine information. In a narrative review by Harrington et al. (2014) examining the role of trust in HPV vaccine uptake, the authors reported that trust in healthcare providers (e.g., medical doctors) was consistently a significant predictor of vaccine uptake [2]. Minorities who mistrusted healthcare providers, government agencies, or pharmaceutical companies were found to have lower rates of vaccine uptake [2]. Furthermore, Harrington et al. (2014) reported that minority groups were more likely to trust HPV vaccine information from their family members and religious organizations in comparison to non-Hispanic white people [2]. In a recent study conducted in a sample of Hispanic people residing along the U.S./Mexico border, several barriers to completing HPV vaccinations emerged, including lacking contact from HCPs, being ineligible for the HPV vaccine, and having medical concerns [3]. Another study that was conducted in a Hispanic-majority sample (89.4%) found that having trust in the government, speaking Spanish, having a larger household size, and having lower levels of perceived community stigma were associated with accepting the HPV vaccine [4]. The current study will extend upon these findings by exploring whom parents trust to provide HPV vaccine information, as well as socio-cultural–behavioral factors that influence parents’ decisions to vaccinate their children with the HPV vaccine. Furthermore, the current study focuses on Hispanic people of Mexican origin to understand the unique barriers that Hispanic individuals face with regard to HPV vaccination.

### Purpose

The purpose of this pilot study was to use a qualitative approach toward obtaining preliminary evidence identifying trustworthy messengers of HPV vaccine information in the eyes of Mexican American (MA) parents of children of 11–17 years of age. Our objective was to conduct focus groups (*N* ≈ 3 groups with 3–5 participants per group) with MA parents of children of 11–17 years of age to identify trustworthy messengers within the community. The focus group findings provide insight and the impetus for designing an intervention tailored to MA parents to promote the HPV vaccine by leveraging trusted messengers of the HPV vaccine. We hypothesized that healthcare providers (HCPs) would be perceived as the most trustworthy sources of information. Specifically, we predicted that medical doctors (MDs), registered nurses (RNs), and pharmacists (PharmDs) would be perceived as the most trustworthy source of HPV vaccine information.

## 2. Materials and Methods

### 2.1. Participants

Three 90 min pilot focus groups, with three to five participants in each group, were conducted. After analyzing the responses from the third focus group, thematic saturation was reached. The inclusion criteria included being 18 years of age or older, residing in El Paso, TX, USA, identifying as MA, speaking English, and being a parent of a child between the ages of 11 and 17. MA parents were recruited online via social media using Facebook’s geo-advertisements specified to El Paso, TX, USA. The advertisement contained a QR code that led participants to a prescreening survey to determine eligibility using the criteria. Eligible participants then provided contact information to schedule a time for the 90 min focus groups, which occurred virtually, using Zoom. The lead investigator then reached out to each participant individually to schedule a time that worked best for the participant. The Zoom platform permitted participants to access the focus groups via two resources: (1) a computer/laptop via a weblink, and (2) a handheld device or tablet via a weblink or by downloading the Zoom application. Permitting parents to participate using Zoom promoted access to participating in focus groups and also accommodated working parents. Two of the focus groups were hosted after working hours while the third was hosted during lunchtime.

### 2.2. Measures

Focus group questions related to general HPV and HPV vaccine inquiry were adapted from Polonijo et al. (2022) [5]. Focus group questions that focused on trusted sources of HPV vaccine information were developed by the investigators. All focus group questions were reviewed by the El Paso Vaccine Promotion Community Advisory Board (EPVP-CAB) for cultural relevance and appropriateness. Table 1 below includes the twelve questions asked during the focus groups.

### 2.3. Study Procedures

The three 90 min focus groups were held over two days via Zoom in June 2022. Prior to recording audio, the participants were presented with an informed consent form approved by The University of Texas at El Paso’s Institutional Review Board (IRB). The focus groups were conducted in English and moderated by the lead investigator of the project. A student research assistant served as a co-facilitator who collected detailed notes and provided feedback to the moderator after each session. All focus groups were audio-recorded using Zoom and later transcribed by the moderator and co-facilitator. In an effort to check reliability, both the investigator and co-facilitator discussed and compared notes immediately after each focus group and then again after transcription was completed. Thematic analysis was used to analyze focus group data and derive themes. Themes were coded manually utilizing transcripts and were discussed and agreed upon. Aligning with ethical guidelines, all participants received a small stipend (USD 50.00) as compensation for their time. Confidentiality was maintained during data storage and analysis.

## 3. Results

### 3.1. Demographics and Background Results

Focus groups were conducted with 15 participants (*M*_age_ = 38.46, *SD* = 5.73; female = 93.3%). Although the sample is predominately female, this gender imbalance may not significantly limit our findings given that mothers often play a dominant role in healthcare decision-making. Table 2 below includes demographics and background information. Additionally, five themes emerged from the focus groups: (1) trusted sources of HPV vaccine information, (2) trust in vaccines, (3) HPV vaccine messaging, (4) vaccination decisions, and (5) knowledge and misconceptions. See Figure 1 below.

### 3.2. Trusted Sources of HPV Vaccine Information

Trust in various healthcare providers was a reoccurring theme throughout all three focus groups. Participants were asked who they trusted the most between doctors, nurses, pharmacists, the CDC, and the WHO. Trust in their child’s pediatrician was consistent across all groups. For example, one participant said, “I trust the pediatrician, she knows best”, while another participant shared, “I agree with the doctor who is the same pediatrician for my child’s whole life, and I trust my doctor”. However, while some parents mentioned that their doctor recommended the vaccine, some shared that theirs had not. For example, one parent stated that, “My doctors made me believe it was not important” and another shared, “my child’s pediatrician never talked about it”.

Trust in registered nurses and pharmacists also emerged. Although most participants agreed that pediatricians were the most trusted source of HPV vaccine information, there was variability in the responses. For example, one participant shared that they trusted nurses more than doctors because they believe that nurses were more involved in patient care in hospital settings. Another participant shared that they had a strong trust in pharmacists with regard to vaccine information because “…they know what it is and how it works and how it’s made”. Importantly, some participants mentioned not having a relationship with their pharmacist, thus impacting their levels of trust in pharmacists. Finally, a few participants mentioned conducting their own research when it came to health information. When probed by the facilitator to share what resources they found to be trustworthy, participants said that they trusted the CDC, the WHO, WebMD, and Mayo Clinics.

### 3.3. Trust in Vaccines

Another theme that emerged was trust in the efficacy of childhood vaccines. The facilitator asked whether parents were for or against vaccinating their children against HPV and why or why not? All 15 participants shared that they trusted vaccines. For example, one parents stated, “I am pro vaccine. I’ll get anything to protect myself or my kids and I’ll be the Guinea pig”. Another shared that they were for vaccines but understood why some parents might be hesitant. They shared, “There is not enough information and people are scared”. Additionally, multiple parents expressed concerns for vaccine side effects, for example, “Severity of side effects are important. HPV vaccine side effects are mild to moderate but if side effects were to be severe, I would be hesitant”. Another parent shared the same sentiment and stated that “Severity of side effects are important because I may not know what could happen in the future”.

### 3.4. HPV Vaccine Messaging

A consistent message conveyed in the focus groups was that the HPV vaccine message itself should focus on cancer prevention rather than STI prevention. For example, one participant stated the following, “It should be marketed as a cancer prevention. STI prevention would not catch my attention because my kids are not having sex”. Another participant shared that, “Before my experiences if it was advertised as an STI vaccine I would not consider it. I might think it is not as important but if advertised as a cancer prevention that would catch my attention”.

### 3.5. Vaccination Decisions

Participants were asked who made the vaccination decisions for their children. As a majority of the participants were mothers, a common theme that emerged from that conversation was that they make the decisions, and their partners usually did not question them. For example, a participant stated that “I am the decision maker and will I share with my husband. If a doctor offered the vaccine, I would do it and then share with my husband”, while another shared, “If I want my kids to get it and I will say that it is a requirement”.

### 3.6. HPV Knowledge and Misconceptions

The final theme that came up in all focus groups was HPV knowledge and misconceptions. Knowledge varied between participants, ranging from hearing about it through their child’s pediatrician to having personal experiences with cervical cancer and HPV. One parent stated, “I thought [the HPV vaccine] was for kids who were sexually active. I found out later that it relates to cancer, and it can also be for boy and my doctor recommended HPV vaccine”. Another parent shared that she had vaccinated her child as soon as they were eligible because her friend had cervical cancer. See Table 3 below.

## 4. Discussion

Several themes emerged in the current study, including trust in healthcare practitioners. Specifically, we found that doctors, nurses, and other healthcare providers were the most trusted sources for HPV vaccine information. Supported findings were reported by Fu et al. (2017), who reported that minorities trusted their healthcare providers most for HPV vaccine information [6]. Another theme that emerged in the current study was trust in the efficacy of childhood vaccines. Supporting this finding, a study by MacArthur (2017) that included adult college students in the U.S. found that many trust the effectiveness of the HPV vaccine in preventing HPV, HPV-related cancers, and genital warts [7]. In addition, there was a theme of vaccination decisions for children heavily relying on parents, with mothers emerging as the primary decision makers. A study conducted by Painter et al. (2019) including Latin American mothers highlighted their parental control and dominance in vaccine decision-making for their children [8]. Encouraging healthcare providers to share HPV vaccine information with parents is of the utmost importance in interventions aimed at increasing HPV vaccine uptake among children. Additionally, targeting parents, specifically mothers, as the primary decision-makers in vaccination decisions is a key strategy for improving HPV vaccine acceptance.

Other themes that emerged in the current study include messaging around the HPV vaccine and how it is promoted and existing knowledge and misconceptions about HPV and the HPV vaccine. Specifically, we found that there was a mutual consensus that HPV messaging should be focused on the vaccine as a cancer-preventative measure instead of a form of STI prevention. Supporting this, a study conducted by Cartmell et al. (2019) found that shifting the focus from STI prevention to cancer prevention would attract the attention of parents and make education on HPV more accessible to parents, particularly for those who believed that their children were not sexually active [9]. Furthermore, another theme that emerged from the current study was the variation in HPV knowledge and misconceptions among the participants. Two separate studies conducted by Albright et al. (2018) and Painter et al. (2019) revealed gaps in participants’ understanding of the HPV virus and vaccine, with the majority lacking comprehension of the term itself or the health consequences associated with the virus [8,10]. Interventions are needed to shift the focus of HPV vaccine messaging towards cancer prevention, to increase HPV vaccine uptake by gaining parental support. Additionally, targeted interventions aimed at increasing knowledge and addressing misconceptions on HPV and the HPV vaccine are crucial in overcoming barriers within the community, creating opportunities for improved vaccine uptake.

There were several strengths to this current study that are important to mention. One notable strength was the inclusion of a Mexican American sample as it addresses a population that may have unique cultural beliefs and practices that influence healthcare decisions. Additionally, the use of focus groups provided valuable insights into MA individuals’ cultural perceptions and needs regarding HPV vaccination. Another strength was the use of Zoom for conducting focus groups, which further ensured the participation of working parents, making the study more inclusive. Lastly, the study was able to identify key themes, such as trust, messaging, vaccination decisions, and HPV knowledge, which can inform future interventions.

It is important to note that there were a few limitations to this current study. First, the study had a sample size of 15 participants in total, which limits the generalizability to the broader population of Mexican Americans. Next, there is the overrepresentation of women, which may misrepresent and underrepresent male parents and caregivers. Future studies should recruit more male participants, to allow us to understand gender-specific perspectives on vaccine decision-making. Another limitation of the current study was that the education level or employment status of the parents was not collected; thus, inferences regarding the impact of education or employment on HPV knowledge could not be identified. A final limitation concerned the recruitment of participants, which was through social media, possibly leading to bias towards those who were more comfortable with using technology and social network websites, potentially excluding parents who were not comfortable or may not have known how to navigate technology or social media.

## 5. Conclusions

In conclusion, the findings from this study have implications for designing health messages and interventions that include trusted messengers for promoting the HPV vaccine, including MDs, RNs, and PharmDs. Specifically, future interventions should leverage trusted healthcare providers and family-centered messaging to enhance vaccine uptake. Moreover, our study suggests that family-centered messages may be effective in promoting the HPV vaccine in MAs. Educational initiatives should focus on the misconceptions identified within this study, including the impact of HPV on boys and girls, given that some participants thought that HPV only impacted girls. Furthermore, interventions that shift the focus of HPV vaccine promotion toward cancer prevention may make the information more relatable and easier to understand. Ultimately, these strategies may create an environment that enhances vaccine acceptance and uptake within the MA community.

## Figures and Tables

**Figure 1 ijerph-22-00069-f001:**
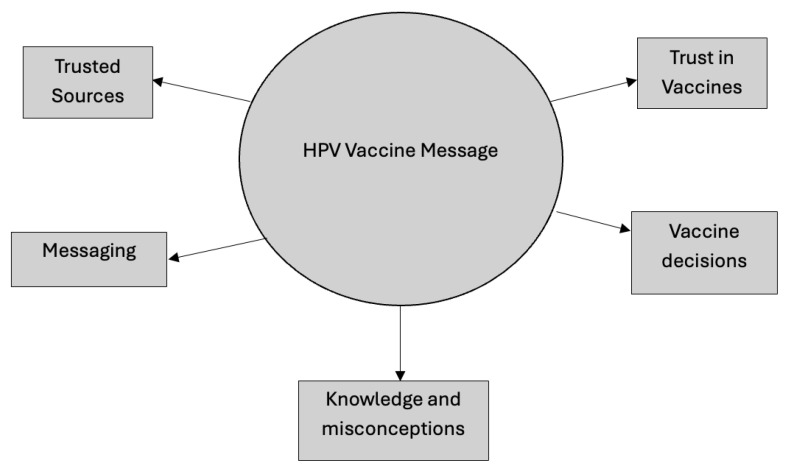
Focus group themes.

**Table 1 ijerph-22-00069-t001:** Focus group questions.

Questions
1. What do you know about Human Papillomavirus (HPV)? What health problems are caused by HPV? 2. Our past research on this topic shows that medical doctors are the most trusted sources of information regarding the HPV vaccine. Would you trust medical doctors for HPV vaccine information? Why or why not? 3. Nurses also emerged as trusted sources of information regarding the HPV vaccine. Would you trust nurses for HPV vaccine information? Why or why not? 4. Would you trust the Centers for Disease and Control (CDC) and the World Health Organization for HPV vaccine information? Why or why not? Which one do you trust more, CDC OR WHO? 5. Pharmacists also emerged as trusted sources of information regarding the HPV vaccine. Would you trust pharmacists for HPV vaccine information? Why or why not? 6. If a message about the HPV vaccine includes the sexually transmitted infection aspect, does that turn you off? In comparison to being a cancer prevention message? 7. What information could help you decide whether or not to get the HPV vaccine for your child? 8. In your family, who makes decisions about your child’s health? 9. If you have a partner, what would your partner think about you vaccinating your child/children against HPV? Given that it prevents cancer? 10. What myths or misconceptions have you heard about HPV and the HPV vaccine? 11. Are you for or against vaccinating your children against HPV? 12. What are some potential barriers do you experience when learning about HPV?

**Table 2 ijerph-22-00069-t002:** Demographics and background information.

**Variable**	**Total Responses**	**Percentage of Responses**
Ethnicity		
Non-Hispanic	1	6.7%
Hispanic/Latinx/Spanish Descent	14	93.3%
Household income		
Less than USD 5000	1	6.7%
USD 5001–USD 20,000	2	13.3%
USD 20,001–USD 40,000	2	13.3%
USD 60,001–USD 80,000	3	20%
USD 80,001–USD 100,000	2	13.3%
USD 100,001 or more	5	33.3%
What is your primary language spoken at home?		
English	13	86.7%
Spanish	2	13.3%
What describes your current relationship status?		
Single	3	20%
Dating	1	6.7%
Married	10	66.7%
Divorced	1	6.7%

**Table 3 ijerph-22-00069-t003:** Visualization of themes and representative quotes.

Theme	Subtheme	Quote
Trusted Sources	Trust in Pediatricians	“I trust the pediatrician, she knows best”.
Trust in Vaccines	Pro-Vaccine	“I am pro vaccine. I’ll get anything to protect my kids and I’ll be the guinea pig”.
HPV Vaccine Messaging	Focus on Cancer Prevention	“It should be marketed as cancer prevention. STI prevention would not catch my attention because my kids are not having sex”.
Vaccination Decisions	Mothers as Decision Makers	“I am the decision maker and will I share with my husband. If a doctor offered the vaccine, I would do it and then share with my husband”.
HPV Knowledge and Misconceptions	General Knowledge of HPV	“I thought [the HPV vaccine] was for kids who were sexually active. I found out later that it relates to cancer, and it can also be for boys and my doctor recommended HPV vaccine”.

## Data Availability

The raw data supporting the conclusions of this article will be made available by the authors on request.

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
