# Peer review of "Exploring Trusted Sources of HPV Vaccine Information Among Mexican American Parents in El Paso, Texas"

_ijerph, 2025, doi:10.3390/ijerph22010069_

Round 1

Reviewer 1 Report

Comments and Suggestions for Authors

Comprehensive Comments and Suggestions for Authors

1. Title and Abstract

  • Title: Consider revising the title to explicitly reflect the target population and geographic focus. For instance, “Exploring Trusted Sources of HPV Vaccine Information Among Mexican American Parents in El Paso, Texas.”
  • Abstract:
    • The abstract is clear but could benefit from including specific findings. For example, mention that "pediatricians, registered nurses, and pharmacists were identified as the most trusted sources of information."
    • Highlight the implications of your findings for public health interventions directly in the abstract.

2. Introduction

  • Strengths: The introduction provides a good overview of the importance of HPV vaccination and the role of trust in vaccine uptake.
  • Suggestions:
    1. Clearly explain why Mexican American parents were chosen as the focus. For instance, discuss the unique barriers or cultural factors this group faces concerning HPV vaccination.
    2. Reference studies that examine trusted sources in other minority populations to provide a broader context and clarify how this study contributes uniquely.
    3. Include more details about the importance of culturally tailored interventions for vaccine promotion, particularly in Hispanic communities.

3. Methods

  • Strengths: Participant recruitment and focus group logistics are described clearly.
  • Suggestions:
    1. Sample Size: Justify why 15 participants across three groups were sufficient to achieve thematic saturation. For example, include a statement such as, “Thematic saturation was reached after analyzing responses from the third focus group.”
    2. Focus Group Questions: Explain how the focus group questions (e.g., those adapted from Polonijo et al., 2022) were tested or validated to ensure cultural relevance and appropriateness for this population.
    3. Thematic Analysis: Describe the specific steps taken during thematic analysis. Did you use software like NVivo or ATLAS.ti for coding? If manual, describe how codes were derived and agreed upon.
    4. Demographics: Provide a rationale for why most participants were female and how this might impact the findings. For example, state, “Given that mothers often play a dominant role in healthcare decision-making, this gender imbalance may not significantly limit our findings.”

4. Results

  • Strengths: The themes are clearly identified and supported by direct quotes.
  • Suggestions:
    1. Visualization: Include a table summarizing themes and representative quotes. For example:

Theme

Subtheme

Representative Quote

Trusted Sources

Trust in Pediatricians

“I trust the pediatrician; she knows best.”

HPV Vaccine Messaging

Focus on Cancer Prevention

“If marketed as cancer prevention, it would catch my attention.”

    1. Discuss variation in responses. For instance, highlight differences in trust levels between healthcare providers like doctors versus pharmacists.
    2. Mention any contradictory responses. For example, while doctors were trusted, some participants noted their pediatricians did not recommend the vaccine.

5. Discussion

  • Strengths: The discussion effectively connects findings to the literature and suggests practical implications.
  • Suggestions:
    1. Provide specific strategies for incorporating trusted messengers into HPV vaccination campaigns. For instance, “Public health campaigns could collaborate with local pediatricians, nurses, and pharmacists to deliver culturally tailored messages in community health centers and schools.”
    2. Reflect on how messaging focused on cancer prevention could be implemented practically. For example, “Promotional materials emphasizing the vaccine’s role in preventing cervical cancer could be distributed at pediatric clinics and pharmacies.”
    3. Address potential limitations of focusing solely on cancer prevention, such as neglecting the STI aspect, which may resonate with other groups.
    4. Discuss implications for broader populations beyond Mexican Americans, indicating areas for future research (e.g., studies targeting fathers or Spanish-speaking-only parents).

6. Conclusions

  • Suggestions:
    1. Add a concise statement recommending next steps for intervention development, such as, “Future interventions should leverage trusted healthcare providers and family-centered messaging to enhance vaccine uptake.”
    2. Highlight the importance of addressing identified misconceptions about the HPV vaccine as part of educational initiatives.

7. Language and Presentation

  • Suggestions:
    1. Revise for grammatical clarity. For example, in the sentence, “...to increase HPV vaccine uptake to ultimately gain the support of parents,” simplify to, “...to increase HPV vaccine uptake by gaining parental support.”
    2. Avoid repetitive phrases like “trusted sources of information” by using synonyms or rephrasing.
    3. Use consistent terminology throughout. For example, instead of alternating between “HPV vaccine information” and “vaccine messaging,” choose one term.

8. Ethical Considerations

  • Suggestions:
    1. Add a brief statement clarifying that monetary compensation was reasonable and not coercive. For example, “Participants received a small stipend ($X) as compensation for their time, aligning with ethical guidelines.”
    2. Confirm that participant confidentiality was maintained during data storage and analysis.

9. Limitations

  • Suggestions:
    1. Expand on the limitation regarding the overrepresentation of female participants by stating, “Future studies should recruit more male participants to understand gender-specific perspectives on vaccine decision-making.”
    2. Acknowledge the potential bias introduced by social media recruitment, such as, “The recruitment method may have excluded parents who are not comfortable with technology or do not use social media.”
    3. Note the limited geographic scope (El Paso, TX) and recommend replication in other regions with diverse Hispanic populations.

10. Figures and Tables

  • Suggestions:
    1. Add a conceptual framework or visual diagram illustrating the trust relationships among healthcare providers, parents, and institutions like the CDC and WHO.
    2. Include a more detailed demographic table showing participants' characteristics, such as age distribution, education level, and employment status.

Summary of Actionable Points

  1. Expand the introduction to better define the study’s contribution and rationale.
  2. Justify sample size and methodological choices in greater detail.
  3. Provide visual aids (e.g., tables and diagrams) to enhance the presentation of results.
  4. Offer more specific and actionable recommendations for public health interventions.
  5. Address limitations more comprehensively, particularly regarding recruitment and generalizability.
  6. Ensure grammatical consistency and reduce repetitive language.

Author Response

Point by Point Response to Reviewer #1

Reviewer 1:  “Consider revising the title to explicitly reflect the target population and geographic focus. For instance, “Exploring Trusted Sources of HPV Vaccine Information Among Mexican American Parents in El Paso, Texas.”

Response to Reviewer 1:   We agree with the reviewers suggestion to modify our title to explicitly reflect the target population and geographic focus.  The revised manuscript now includes the reviewer’s suggested title.  “Exploring Trusted Sources of HPV Vaccine Information Among Mexican American Parents in El Paso, Texas”

Reviewer 1:  “The abstract is clear but could benefit from including specific findings. For example, mention that "pediatricians, registered nurses, and pharmacists were identified as the most trusted sources of information."  Highlight the implications of your findings for public health interventions directly in the abstract.

Response to Reviewer 1:  Reviewer #1 notes that the abstract could benefit from including specific findings.  We agree with the reviewers comment and the revised manuscript now address this in the abstract on page 1, lines 19-20.  Moreover, the reviewer suggests that we should highlight the implications of our findings in the abstract.  We appreciated this suggestion and have revised the abstract to now state the following, " Findings from this study have implications for designing public health interventions that leverages pediatricians, registered nurses, and pharmacists to promote the HPV vaccine in parents.”  See page 1, lines 20-22.

Reviewer 1:  Introduction:  Clearly explain why Mexican American parents were chosen as the focus. For instance, discuss the unique barriers or cultural factors this group faces concerning HPV vaccination.  Reference studies that examine trusted sources in other minority populations to provide a broader context and clarify how this study contributes uniquely.  Include more details about the importance of culturally tailored interventions for vaccine promotion, particularly in Hispanic communities.

Response to Reviewer 1:  The reviewer suggested that we clearly explain why we focused on Mexican American parents.  The revised manuscript now includes literature in the area of trusted sources of HPV information in Hispanics and the unique barriers that Hispanics face toward HPV vaccination.  Specifically, on pages 1-2, lines 44-55, we write the following, “In a recent study conducted in a sample of Hispanics residing along the U.S./Mexico border, several barriers to completing HPV vaccinations emerged including lacking contact from HCPs, being ineligible for the HPV vaccine, and having medical concerns [3]. Another study that was conducted in a Hispanic-majority sample (89.4%) found that having trust in the government, speaking Spanish, having a larger household size, and having lower levels of perceived community stigma were associated with accepting the HPV vaccine [4]. The current study will extend upon these findings by exploring who parents trust to recommend HPV vaccine information, as well as socio-cultural-behavioral factors that influence parents decisions to vaccinate their children with the HPV vaccine. Furthermore, the current study focuses on Hispanics of Mexican origin to understand the unique barriers that Hispanics face with HPV vaccination.”

Reviewer 1:  Sample Size: Justify why 15 participants across three groups were sufficient to achieve thematic saturation. For example, include a statement such as, “Thematic saturation was reached after analyzing responses from the third focus group.”

Response to Reviewer 1:  The revised manuscript now included the reviewers suggested language.  Our team did in fact reach saturation in the third focus group and realized that we failed to include this statement initially, thus, we appreciate the reviewers recommended insert. 

Reviewer 1:  Focus Group Questions: Explain how the focus group questions (e.g., those adapted from Polonijo et al., 2022) were tested or validated to ensure cultural relevance and appropriateness for this population.

Response to Reviewer 1:  Our team failed to include this in the initial version of the manuscript.  The revised manuscript now states on page 2, lines 88-90, “All focus group questions were reviewed by the El Paso Vaccine Promotion Community Advisory Board (EPVP-CAB) for cultural relevance and appropriateness.” 

Reviewer 1:  Thematic Analysis: Describe the specific steps taken during thematic analysis. Did you use software like NVivo or ATLAS.ti for coding? If manual, describe how codes were derived and agreed upon.

Response to Reviewer 1:  The revised manuscript now includes the following on page 3, lines 100-104, “In an effort to check for reliability, both the investigator and co-facilitator discussed and compared notes immediately after each focus group and then again after transcription was completed. Thematic analysis was used to analyze focus group data and derive themes. Themes were coded manually utilizing transcripts and were discussed and agreed upon.”

Reviewer 1:  Demographics: Provide a rationale for why most participants were female and how this might impact the findings. For example, state, “Given that mothers often play a dominant role in healthcare decision-making, this gender imbalance may not significantly limit our findings.”

Response to Reviewer 1:  The revised manuscript now includes the suggested language on page 3, lines 110-112.  Specifically, we write the following, “Although the sample is predominately female, this gender imbalance may not significantly limit our findings given that mothers often play a dominant role in healthcare decision-making.”

Reviewer 1: Visualization: Include a table summarizing themes and representative quotes. For example:

Theme

Subtheme

Representative Quote

Trusted Sources

Trust in Pediatricians

“I trust the pediatrician; she knows best.”

HPV    Vaccine Messaging

Focus  on        Cancer Prevention

“If marketed as cancer prevention, it would catch my attention.”

Response to Reviewer 1:  The revised manuscript now includes the suggested table on page 5, line 163. 

Reviewer 1: Discuss variation in responses. For instance, highlight differences in trust levels between healthcare providers like doctors versus pharmacists. Mention any contradictory responses. For example, while doctors were trusted, some participants noted their pediatricians did not recommend the vaccine.

Response to Reviewer 1: We appreciate the reviewer’s suggestion and the revised manuscript now addresses this issue on page 4, lines 128-138.  Specifically, we write the following, Trust in registered nurses and pharmacists also emerged. Although most participants agreed that pediatricians were the most trusted source of HPV vaccine information, there was variability in responses. For example, one participant shared that they trusted nurses more than doctors because they believe that nurses are more involved in patient care in hospital settings. Another participant shared that they had strong trust in pharmacists in regard to vaccine information because “…they know what it is and how it works and how it’s made”. Importantly, some participants mentioned not having a relationship with their pharmacist, thus impacting their levels of trust in pharmacists. Finally, a few participants mentioned doing their own research when it comes to health information. When probed by the facilitator to share what resources they found to be trustworthy, participants said they trust the CDC, WHO, WebMD, and Mayo Clinic.

Reviewer 1.  Conclusions:  Add a concise statement recommending next steps for intervention development, such as, “Future interventions should leverage trusted healthcare providers and family-centered messaging to enhance vaccine uptake.”

Response to reviewer 1:  We appreciate the reviewers suggested statement for recommending next steps for interventions.  The revised manuscript includes the suggested language on page 7, lines 234-237.  

Reviewer 1.  Highlight the importance of addressing identified misconceptions about the HPV vaccine as part of educational initiatives.

Response to reviewer 1:  We appreciate the reviewers suggestion and we now include the following on page 7, 237-239, “Educational initiative should focus on the misconceptions identified within this study including the impact of HPV on boys and girls, given that some participants thought HPV only impacted girls.”

 Reviewer 1 “Revise for grammatical clarity. For example, in the sentence, “...to increase HPV vaccine uptake to ultimately gain the support of parents,” simplify to, “...to increase HPV vaccine uptake by gaining parental support. Avoid repetitive phrases like “trusted sources of information” by using synonyms or rephrasing.  Use consistent terminology throughout. For example, instead of alternating between “HPV vaccine information” and “vaccine messaging,” choose one term.”

Response to reviewer 1:  The revised manuscript has been proofread and edited to improve grammatical clarity.  The reviewer also suggested that we avoid repetitive phrased like “Trusted sources of information” by using synonyms.  Importantly, the revised manuscript still includes the use of “Trusted sources of information” throughout the manuscript given that the literature utilizes this phrase in the area of research and we wanted to remain consistent and reduce confusion.

 Reviewer 1. Add a brief statement clarifying that monetary compensation was reasonable and not coercive. For example, “Participants received a small stipend ($X) as compensation for their time, aligning with ethical guidelines.”

Response to reviewer 1:  The revised manuscript now includes the suggested wording on page 3, 105-106. 

Reviewer 1.  Confirm that participant confidentiality was maintained during data storage and analysis.

Response to reviewer 1:  The revised manuscript now includes this language on page 3, line 106. 

Reviewer 1.  Expand on the limitation regarding the overrepresentation of female participants by stating, “Future studies should recruit more male participants to understand gender-specific perspectives on vaccine decision-making.” Acknowledge the potential bias introduced by social media recruitment, such as, “The recruitment method may have excluded parents who are not comfortable with technology or do not use social media.” Note the limited geographic scope (El Paso, TX) and recommend replication in other regions with diverse Hispanic populations.

Response to reviewer 1:  The revised manuscript now addresses all of the reviewers suggestions on page 7, lines 224-232. 

Reviewer 1 Add a conceptual framework or visual diagram illustrating the trust relationships among healthcare providers, parents, and institutions like the CDC and WHO.  Include a more detailed demographic table showing participants' characteristics, such as age distribution, education level, and employment status.

Response to reviewer 1:  We appreciate the reviewer’s suggestions to include a visual framework illustrating the trust relationships and also including a more detailed demographic table showing participants’ characteristics including age distribution, education level, and employment status.  Notably, we did not assess education level or employment status and the revised manuscript now explicitly mentions this limitation on page 7, lines 224-232. We decided to not include age within table 2 given that table 2 provides frequencies and percentages, moreover, we provide the age and standard deviation of our sample of page 3, line 109.  Regarding the visual framework, our team agrees that a visual diagram illustrating the trust relationships among healthcare providers would be impactful, however, a quantitative approach may be more appropriate to include effect size indices for each source of trust. However, if the reviewer believes that it is imperative that we include this diagram, our team is willing to develop it. 

Reviewer 2 Report

Comments and Suggestions for Authors

1. The main question addressed by the research was to identify 13 trustworthy messengers of HPV vaccine information in Mexican American (MA) parents of children of 11–17 years of age.

2.  I consider the topic original as it addresses a specific gap and in a specific population of Mexican American parents in the field administering HPV to prevent Cervical cancer as it has a high prevalence.

3. The question addressed allows permitted parents to participate in the study using Zoom which promotes access to participating in focus groups and also accommodated working parents. More to that two of the focus groups were hosted after working hours while the third was hosted during lunchtime. In other studies researchers used focus groups at work and this was inconvenient.

4. The specific improvements the authors could consider regarding the methodology would have been to include level of education of the parents to correlate with information about the HPV.

5. I agree with the authors conclusions for there are consistent with the evidence and arguments presented and the conclusions address the main question posed and ultimately, these strategies suggested may create an environment where it enhances vaccine acceptance and uptake within the MA community.

6.   The author referred to appropriate references for there are UpToDate.

7.  An additional comment on the table 2 as already stated the author could have added “educational level of the participants.”

Author Response

Point by Point Response to Reviewer #2

Reviewer 2:  “The specific improvements the authors could consider regarding the methodology would have been to include level of education of the parents to correlate with information about the HPV.”

Response to Reviewer #2:   We appreciate the reviewer’s suggested improvement to include level of education of the parents to correlate with information about HPV.  We agree that this information would have strengthened the quality of our findings by providing insight into the impact that education has on HPV information and who individuals trust for HPV information.  We have now addressed this issue on page 7, lines 226-228.  The edits are highlighted in the manuscript. 

Reviewer 2:  7.  An additional comment on the table 2 as already stated the author could have added “educational level of the participants.”

Response to Reviewer 2:  Reviewer #2 notes that table 2 could have included education level.  We agree and in future studies will ensure to assess education level.  We have noted the lack of assessing education level as a limitation on page 7, lines 226-228.